# Site-Specific Considerations on Engineered T Cells for Malignant Gliomas

**DOI:** 10.3390/biomedicines10071738

**Published:** 2022-07-19

**Authors:** Nirmeen Elmadany, Obada T. Alhalabi, Michael Platten, Lukas Bunse

**Affiliations:** 1DKTK (German Cancer Consortium), Clinical Cooperation Unit (CCU), Neuroimmunology and Brain Tumor Immunology, German Cancer Research Center (DKFZ), 69120 Heidelberg, Germany; nirmeen.elmadany@dkfz-heidelberg.de (N.E.); m.platten@dkfz-heidelberg.de (M.P.); 2Department of Neurology, Medical Faculty Mannheim, MCTN, University of Heidelberg, 68167 Mannheim, Germany; 3Brain Tumor Translational Targets, DKFZ Junior Group, German Cancer Research Center (DKFZ), 69120 Heidelberg, Germany; o.alhalabi@dkfz-heidelberg.de; 4Department of Neurosurgery, Heidelberg University Hospital, 69120 Heidelberg, Germany; 5Immune Monitoring Unit, National Center for Tumor Diseases (NCT), 69120 Heidelberg, Germany; 6Helmholtz Institute of Translational Oncology (HI-TRON), 55131 Mainz, Germany; 7DKFZ Hector Cancer Institute, University Medical Center Mannheim, 68167 Mannheim, Germany

**Keywords:** CARs, CAR T cells, glioma, glioblastoma, intrathecal delivery, systemic delivery, FUS, nanotechnology

## Abstract

Immunotherapy has revolutionized cancer treatment. Despite the recent advances in immunotherapeutic approaches for several tumor entities, limited response has been observed in malignant gliomas, including glioblastoma (GBM). Conversely, one of the emerging immunotherapeutic modalities is chimeric antigen receptors (CAR) T cell therapy, which demonstrated promising clinical responses in other solid tumors. Current pre-clinical and interventional clinical studies suggest improved efficacy when CAR-T cells are delivered locoregionally, rather than intravenously. In this review, we summarize possible CAR-T cell administration routes including locoregional therapy, systemic administration with and without focused ultrasound, direct intra-arterial drug delivery and nanoparticle-enhanced delivery in glioma. Moreover, we discuss published as well as ongoing and planned clinical trials involving CAR-T cell therapy in malignant glioma. With increasing neoadjuvant and/or adjuvant combinatorial immunotherapeutic concepts and modalities with specific modes of action for malignant glioma, selection of administration routes becomes increasingly important.

## 1. Introduction

Malignant gliomas represent a heterogenous group of aggressive tumors originating in the central nervous system (CNS) and comprising up to 80% of primary brain tumor cases [1]. The most common malignant glioma is glioblastoma (GBM), which accounts for around 48% of all CNS cancers [2]. Despite tremendous advances in biotechnology, genomics and, thus, precision medicine, GBM patient prognosis remains poor with a median overall survival (OS) ranging from 14 to 16 months [3]. The current standard of care (SOC) includes surgical resection followed by adjuvant temozolomide (TMZ) and radiotherapy (RT) [4]. However, recurrence is still inevitable following SOC and viable treatment options remain scare [5]. Therefore, there is an urgent need for novel therapeutic modalities to improve patient prognosis and survival [6]. The immunosuppressive (Box 1) and hypoxic microenvironment of GBM represents one of the key factors that drive resistance to immunotherapy mechanistically leading to T cell exhaustion. The development of immune checkpoint inhibitors (ICI) posed a breakthrough in the therapy of various solid tumors, such as melanoma or non-small cell lung cancer. However, they have not yet conferred clinical benefit for glioma patients [7]. The limited immune cell trafficking across the blood-brain-barrier (BBB) in GBM and the low mutational load of 30–50 non-synonymous mutations in GBM hinder a substantial response to such immunotherapeutic modalities [8,9]. Data from phase III clinical trials (CheckMate-143 and Checkmate-498; NCT02617589) and the preliminary results of an ongoing complementary trial (CheckMate-548, NCT02667587) showed no ICI-related extension of GBM patient survival [10,11]. Nevertheless, the phase I trial NCT02313272 exploring the effect of combining immunotherapy with radiotherapy showed improvement in patient progression-free survival (PFS) [12,13]. Currently, several ongoing trials are exploring the combination of ICI with other therapies in patients with newly diagnosed and recurrent GBM, for example with anti-angiogenic factors such as bevazicumab in the phase II trial NCT02337491 [11]. Furthermore, the ICI pembrolizumab which is a programmed cell death protein 1 (PD-1) inhibitor was found to significantly extend the overall survival as a neoadjuvant therapy in a randomized, open-label, pilot multi-institution clinical trial by the Ivy Foundation Early Phase Clinical Trials Consortium [14]. In line with this finding, Schalper et al. 2019 demonstrated that a neoadjuvant therapy of the PD-1 inhibitor nivolumab activates GBM immune microenvironment in a single-arm phase II clinical trial (NCT02550249) [15].

Adoptive T cell therapy (ACT) is another emerging immunotherapeutic modality aimed at pervading active cytotoxic T cells into the tumor microenvironment [16]. ACT encompasses tumor-infiltrating lymphocyte (TIL) transfer, T cell receptor (TCR) T cell therapy and chimeric antigen receptor (CAR). Although adoptively transferred TILs from melanoma brain metastasis were in principle capable of crossing the BBB, metastasis further progressed following ACT with TIL [17,18]. Therefore, the authors of the study suggested the assessment of locoregional administration of TIL [18]. Nevertheless, in one of the early phase I trials, intrathecal administration of interleukin 2 (IL-2)-stimulated lymphocytes via a catheter led to neurological deterioration and irreversible toxicities in some patients [19]. Although the local administration of TIL seems to overcome the restricted BBB accessibility, the accompanied side effects might be attributed to the low specificity of IL-2 expanded lymphocytes [19,20]. Hence, highly specific ACT against tumor-associated antigens and preferentially neoepitopes are critically needed.

TCR-T cell therapy implies the interaction between T cells harboring genetically modified TCR and specific tumor targets presented by the major histocompatibility complex (MHC) molecules on the tumor cell surface [21,22]. An efficacious and safe TCR-T cell therapy requires a selective binding between tumor neoantigens which are absent in healthy tissue and presented by tumor MHC, and the cognate TCR binding partners [23,24]. Nowadays, the emerging advances and accessibility of TCR sequencing facilitate TCR clonotype and repertoire profiling which help in identifying tumor antigen-specific TCRs [25,26]. Nevertheless, about 40–90% of known tumors are considered MHC class I (MHCI) deficient and thereby express low immunogenicity [27,28].

CAR-T cells are engineered to express recombinant receptors which are capable of recognizing specific tumor antigens [20] (Box 2). The antigen-recognition moieties of CARs endue the engineered autologous T cell with MHC–independent reactivity against the tumor. Five generations of CAR-T cells have evolved (Box 2), each new generation developed advances on the CAR construct to increase the specificity and avert off-target toxicity [20,29,30]. In preclinical studies, several CARs against glioma-associated targets have been developed [31].

Six CAR-T-cell therapies have been currently approved by the Food and Drug Administration (FDA), mainly for the treatment of hematological cancers, however, CAR-T cell therapies in brain tumors are not included [32]. In malignant gliomas, several clinical trials targeting disialgoganglioside 2 (GD2) [33], interleukin 13 receptor subunit alpha 2 (IL13Ra2), human epidermal growth factor receptor 2 (Her2), B7 homolog 3 (B7-H3), epidermal growth factor receptor variant III (EGFRvIII), CD133, CD147, ephrin type-A receptor 2 (EphA2) or chlorotoxin [31], are recruiting or have been conducted [31]. However, primary or secondary resistance is caused by tumor heterogeneity, antigen loss and limited intratumoral CAR-T cell abundance [34]. CAR-T cell migration across the BBB is still a critical hurdle that limits the efficacy of targeted CAR-T cell therapies in glioma. Indeed, locoregional delivery of CAR-T cells holds promise in the treatment of malignant gliomas [35].

In this review, we discuss possible routes of administration for CAR-T cells in malignant gliomas (Figure 1), highlighting the most efficacious and safe routes in preclinical experiments as well as in clinical trials to secure sufficient glioma T cell homing. We also shed light on current and planned trials exploring different application routes and glioma antigens.

## 2. Routes of CAR-T Cells Administration in Glioma

### 2.1. Locoregional Delivery of CAR-T Cells in Glioma

#### 2.1.1. Intraventricular Injection

Historically, the CNS has been considered an immune-privileged organ which lacks the conventional lymphatic drainage system. This concept has evolved, with a constant immune surveillance mechanism within the meningeal compartment described in several studies [36,37]. Louveau et al. demonstrated in 2015 that functional lymphatic endothelial cells line the dural sinuses and assemble to form lymphatic vessels to transfer immune cells from the cerebrospinal fluid (CSF) [38]. The discovery of meningeal lymphatic vessels unveiled the mystery behind the excessive influx of adaptive immune cells in neuroinflammatory diseases, such as multiple sclerosis (MS) [39,40,41]. Indeed, preclinical murine-based data showed regression of the T cell-mediated inflammatory response in experimental autoimmune encephalomyelitis (EAE) after the ablation of meningeal lymphatic vessels [39]. Hence, taking advantage of this functional gateway to promote CAR-T cell homing into the brain tumor seemed a promising strategy. Intracerebroventricular (i.c.v.) delivery of EPHA2-, HER2- and IL13Rα2-targeted trivalent CAR-T cell therapy in patient-derived xenograft (PDX) mouse models of posterior fossa ependymoma and metastatic medulloblastoma revealed an increase in the survival of two out of three PDXs following a single i.c.v. dose of the trivalent CAR-T cells. Interestingly, repeating i.c.v. administrations of EPHA2 CAR-T cells in medulloblastoma PDX models enhanced the overall survival in comparison to a single EPHA2 CAR-T cell treatment with no signs of severe systemic toxicities [42].

In a first-in-human phase I clinical trial (NCT04196413), GD2-directed CAR-T cells were applied i.c.v. in patients with diffuse intrinsic pontine glioma (DIPG) who showed partial clinical benefit after a single i.v. dose of 1 × 10^6^/kg GD2-CAR-T cells [33] (Table 1). No CAR-T cell on-target off-tumor toxicity was observed, rather, an activation of the immunosuppressed glioma microenvironment and elevation of proinflammatory cytokines and chemokines as CXCL9, CCL2, TNF-α and IFN-γ was described. Furthermore, radiographic and clinical improvement were observed in three out of four patients [33].

#### 2.1.2. Intratumoral and Intracavitary CAR-T Cell Delivery

Local injection into the tumor is mainly applied in preclinical studies [43,44], whereas delivery of CAR-T cells into the surgical cavity has been utilized in several human clinical trials [10,45,46]. In a preclinical study, Priceman et al. 2018 demonstrated that experimental breast cancer brain metastasis responded similarly in terms of tumor-infiltration and extended preclinical survival after i.c.v. or intratumoral locoregional administration of 0.5 × 10^6^ HER2-directed BBζ CAR-T in BBM1 tumor-bearing mice [47]. Notably, earlier responses have been observed upon direct intratumoral delivery in comparison to the i.c.v. route which might be due to the trafficking time that is required for the CAR-T cells to migrate from the ventricle to the tumor tissue. However, the strict intratumoral delivery might be less advantageous compared to the regional i.c.v. application in multifocal brain metastases, depending on individual cases and affected regions [47,48]. In a primary central nervous system lymphoma (PCNSL) mouse model treated with CD19-directed CAR-T cells either by intratumoral or i.v. injection, a successful eradication of the large established tumor was observed in the intratumoral, but not the i.v. route of administration [49]. Similarly, compared to systemic delivery, improved preclinical OS was observed when B7H3-directed CAR-T cells targeting orthotopic atypical teratoid/rhabdoid tumor xenografts were administered locoregionally [50]. By employing bioluminescence imaging and NanoLuc luciferase-expressing CARs, B7H3-directed CAR-T cells were detected within the tumor five days after either i.t. or i.c.v. injections but not after i.v. injection using tenfold the dose [50]. Despite promising results in preclinical trials using direct intra-tumoral injection, further clinical trials are needed for a conclusive demonstration regarding the efficacy and the injection-site toxicity in humans [10,51].

Direct CAR-T cell infusion into the resection cavity was first described in the first-in-human pilot trial (NCT00730613). Brown et al. demonstrated the safety of up to 12 adjuvant infusions of IL13Rα2-directed CAR-T cells into the cavity in three patients with recurrent GBM at a maximum dose of 10^8^ cells [45]. Indeed, the therapy was well-tolerated with mild side effects such as headaches and transient non-localizing neurologic deficits. Although an increased necrotic tumor volume was observed in brain imaging, the three patients developed recurrent glioma [45]. In the ongoing trial NCT02208362 the intratumoral route of administration has been switched to i.c.v. [52]. Whereas multiple infusions of 10 × 10^6^ IL13Rα2-directed CAR-T cells were administered into the resection cavity in a patient with recurrent GBM, i.c.v. infusion at a late stage of multifocal tumor recurrence caused a regression of intracranial and spinal manifestations with a persistent clinical response for 7.5 months [46]. However, the single observation limits the reliability of final conclusions, and results of the ongoing trial are yet to be published (Table 1).

### 2.2. Systemic Administration of CAR-T Cells

#### Intravenous CAR-T Cells Delivery

Preclinical data has shown limited brain-accessibility of CAR-T cells and the subsequent tumor response following i.v. injection into the mouse tail veins [47,48,51,52]. A further arm in the aforementioned study by Priceman et al. comprised the i.v. application of HER2-directed BBζ CAR-T cells in a mouse model of human breast cancer brain metastasis. Compared to the intracranial administration group, i.v. injection of an equivalent CAR-T cell dose showed limited efficacy with a partial response even after 10-fold escalation [47]. This finding was further confirmed by Brown et al. in a recurrent GBM mouse model treated with HER2-directed BBζ CAR-T cells [52].

Demonstrating similar effects in the context of other tumor antigens, Agliardi et al. generated a CAR construct harboring the murine CD28-derived transmembrane domain and CD3ζ intracellular domains in addition to a single chain variable fragment derived from the EGFRvIII-specific MR1 antibody [50]. The investigators demonstrated that i.v. injection of the EGFRvIII-directed CAR-T cells in orthotopic EGFRvIII^+^ GBM mouse model failed to show any significant response of the established tumor. However, the concomitant intratumoral administration of the pro-inflammatory cytokine IL-12 following lymphodepletion by total body irradiation (TBI) improved the efficacy of the CAR-T cells in terms of tumor size and overall survival [48]. Lymphodepletion before i.v. administration of CAR-T cells was further applied in several studies to enhance CAR-T cell efficacy. Rousso-Noori et al. injected GBM-bearing mice with a single dose of 200 mg/kg of cyclophosphamide intraperitoneally (i.p.) one day before the i.v. administration of 4 × 10^6^ p32-specific mCAR-T cells [52]. A second dose was given on the next day. The median survival of the treated arm was 73 days compared to 41 days of the control group with 30% of the mice remained tumor-free until the end of the experiment [51].

In 2017, O’Rourke et al. published the first-in-human clinical trial NCT02209376 in which up to 5 × 10^8^ EGFRvIII-directed CAR-T cells were administered to ten GBM patients i.v. [53]. Surgical resection in 7 out of the 10 patients confirmed the presence of CAR-T cells in situ within the tumor. However, EGFRvIII antigen loss and immune-escape mechanisms were also detected. Clinically significant neurologic events were observed in three patients [53]. A second clinical trial (NCT01454596) on EGFRvIII-targeted CAR-T cell therapy was published by Goff et al.; aiming at exploring safety and PFS after CAR-T cell therapy [54]. In this study, prior chemotherapy-induced lymphodepletion was performed with cyclophosphamide and fludarabine in 18 GBM patients. Next, up to 2.6 × 10^10^ CAR-T cells were administered i.v. Two patients developed acute dyspnea shortly after the CAR-T cell therapy which was explained by the investigator as dose-dependent T cell-induced congestion of the pulmonary vessels [54]. Despite prior lymphodepletion and post-transfer treatment with IL-2, median PFS was 1.3 months with no objective responses; suggesting limited efficacy and off-tumor toxicity of the systemic i.v. administration of CAR-T cells [54].

Trials on other cancer entities unveiled some side effects which were associated with the systemic i.v. administration of CAR-T cells including the immune effector cell–associated neurotoxicity syndrome (ICANS), cytokine release syndrome (CRS) and on-target-off-tumor toxicities [55,56]. It still not clear whether alternative local delivery routes would evade, develop, or enhance such side effects.

### 2.3. Intra-Arterial CAR-T Cell Delivery

Intra-arterial (i.a.) drug delivery has been used since 1964 for patients with brain tumor mainly to deliver chemotherapeutic drugs through the femoral artery or through a catheter inserted into the organ-feeding arteries to allow repeated administration of a therapeutic [57,58]. Depending on the chosen artery, i.a. application could be regarded as a regional route of administration which might increase the tumor drug-uptake and thereby improve tumor response. With regards to CAR-T cell delivery, limited data is available for malignant gliomas. Nevertheless, i.a. administration in other cancer entities as in liver metastases was well-tolerated with no instances of CRS or other systemic toxicities. Moreover, a favorable biological response, including higher pro-inflammatory cytokines and lower rates of immune exhaustion was demonstrated [59,60,61]. In a preclinical study, activated TILs expanded from patient GBM were successfully isolated from a rabbit brain following i.a. injection into the internal carotid artery. The authors demonstrated the safety of the route since no catastrophic embolic–ischemic events were observed [62], suggesting that i.a. CAR-T cell delivery could be well-tolerated and effective. Preclinical studies are needed to compare the safety and efficacy of this route of administration with front line locoregional CAR-T cell delivery outlined earlier. Once a well-tolerated prime efficacy of i.a. CAR-T therapy is demonstrated, further clinical trials are still needed to confirm the patient safety and determine clinical response.

**Table 1 biomedicines-10-01738-t001:** Published studies involving CAR-T cell therapies in malignant gliomas.

Entity	Patients Enrolled	Application	CAR	Additional Therapies	Outcome	Adverse Events	References
Recurrent glioblastoma Case report	*n* = 1	Locoregional	IL13Rα2	Surgical resection, RT, TMZ, Carmustine, Bevazicumab	Assessment of detection with 18F-FHBG PET	No adverse events reported	[63]
Recurrent glioblastoma	*n* = 3	Locoregional	IL13Rα2	Surgical resection, RT, TMZ, Carmustine, Bevazicumab	Response in 2 patients Increased necrotic tumor volume Therapy driven antigen loss	Grade 3 headache, transient neurology	[45] NCT00730613
Recurrent multifocal glioblastoma Case report	*n* = 1	Locoregional (Intracavitary and intracerebroventricular)	IL13Rα2 4-1BB costimulatory domain and mutated IgG4-Fc linker	Surgical resection, RT, TMZ	Initial clinical response with regression of all intracranial and spinal tumors for 7.5 months was observed	No toxic effects observed	[46]
Newly diagnosed/Recurrent glioblastoma Phase I	*n* = 11	Systemic	EGFRvIII	Surgical resection RT, TMZ, Lomustine, Carboplatin, DCVax(^®^)-L	Early termination No tumor regression observed. Median OS 8 Months Immunoediting with Treg cell abundance and antigen loss. Successful brain trafficking	No serious events	[53] NCT02209376
Recurrent glioblastoma Phase I/II	*n* = 18	Systemic with supportive i.v. interleukin-2	3rd-gen EGFRvIII- CD28 and 4-1BB costimulatory domains	Surgical resection RT, TMZ, Bevazicumab Co-therapy with cyclophosphamide, fludarabine and aldesleukin	No meaningful response OS 6.9 months	Respiratory events with severe hypoxia at highest dose	[54] NCT01454596
Recurrent glioblastoma Phase I	*n* = 18	Systemic	HER2	Many surgical resections RT, TMZ	8 patients showed a clinical benefit, others no response	No serious adverse events recorded	[64] NCT01109095
Anaplastic astrocytoma/metastasized ependymoma (WHO grade III)	*n* = 3	Locoregional	HER2	Surgical resection, chemotherapy	1 stable disease 2 progressive disease CSF detection of immune activity	Neurological worsening after infusion, no further adverse events	[65] Interim analysis NCT03500991
Recurrent glioblastoma	*n* = 1	Locoregional	B7-H3	Surgical resection RT, TMZ	Initial clinical response per MRI	Headache, altered consciousness Patient dropped after 7 cycles	[66]
DIPG and H3K27M-mutated diffuse midline gliomas	*n* = 4	Intravenous and Locoregional	retro-viral expressing GD2-CAR	Surgical resection, RT Lymphodepletion	3 of 4 patients showed neurological and MRI improvement	No on-target off-tumor toxicity	[33,67] NCT04196413

DIPG: diffuse intrinsic pontine gliomas; RT: Radiotherapy; TMZ: Temozolomide; 18F-FHBG PET: 9-(4-(18)F-Fluoro-3-[hydroxymethyl]butyl)guanine ((18)F-FHBG) Positron Emission Tomography.

### 2.4. Focused Ultrasound-Aided CAR-T Cell Homing

Focused ultrasound (FUS) is an emerging innovative tool with multiple clinical applications, including BBB disruption with microbubbles for a better drug delivery, neurocircuit modulation, immune cell modulation and thermo-ablation [68,69,70,71,72]. As previously highlighted, one of the major challenges that faces CAR-T cell therapy is the limited trafficking across the BBB. Indeed, around 98% of small molecules are incapable of passing the BBB to establish a sufficient drug level in brain tissue [73]. The principle of FUS action consists of a transient increase in the permeability of the BBB by the systemic injection of microbubbles followed by a site-specific low-energy burst-tone FUS [74,75]. The microbubbles establish physical cavitations that sonicate upon an externally-applied ultrasonic pulse and permit the transient opening of BBB tight junctions while preserving neuron viability [76]. Being a growing field at an early stage, limited preclinical and clinical data about the effect of FUS on CAR-T cell delivery in malignant glioma is available. However, several clinical trials are currently exploring the safety and feasibility of FUS in malignant glioma. Conceptually, especially in non-enhancing gliomas such as oligodendrogliomas and astrocytomas that are signified by immune cell-excluding tumor microenvironments, further studies combining immune interventions and FUS are required. In one of the preclinical studies, Sabbagh et al. 2021 demonstrated an enhancement of EGFRvIII-directed CAR-T Cell delivery in EGFRvIII-U87-tumor-bearing NSG mice using Low-intensity pulsed ultrasound (LIPU) with a better survival of 129% compared to CAR-T cell therapy alone [77].

Interestingly, Wu et al. 2021 employed the FUS-induced heat to control the activation period of CAR-T cells harboring heat-inducible genes in a mouse model of prostate tumor. Focal administration of FUS activates the Cre–loxP mediated FUS-inducible CAR-T cells only at the tumor site but not in other brain regions which are not exposed to FUS to reduce non-target effects. Furthermore, once the FUS pulses are switched off, the activity of CAR-T cells subsides gradually. Applying this strategy, the authors demonstrated significant tumor regression in the tumor-bearing mice [72]. Such an approach might control the on-target off-tumor toxicities and it would be interested to explore it in malignant glioma mouse model.

### 2.5. Outlook on Biomaterial Carriers and Nano-Technology-Based Delivery

One of the promising strategies to ensure sufficient CAR-T cell delivery into the tumor site is the direct intra-tumoral implantation of a catheter or a biopolymer. More interestingly, this approach can be employed to manage in-operable tumors by placing the implant near the tumor site. Application of this route to deliver CAR-T cell therapy in malignant gliomas has not been established yet. However, it has been successfully tested in other tumor types. Stephan et al. 2015 applied bioactive polymer implant to deliver, expand and disperse TILs in a multifocal ovarian cancer model which could successfully suppress tumor growth, reduce tumor relapse and enhance survival with no serious toxicities [78]. Moreover, the authors demonstrated improved responses to the scaffold-delivered TILs compared to intravenous and direct resection bed delivery, suggesting an effective delivery of CAR-T cells by this carrier-aided route. Indeed, Smith et al. 2017 successfully employed the same polymer to deliver CAR-T cells in orthotopic mouse models of melanoma and pancreatic cancer [79]. The investigators observed increased influx of the NKG2D-directed CAR-T cells co-expressing CBR-luc in vivo from the biopolymer scaffold into the tumor tissue compared to systemic administration. Furthermore, unlike the systemic route, the implanted CAR-T cells permitted robust tumor regression, activated endogenous tumor reactive lymphocytes and antigen presenting cells (APC), elicited antitumor immunity and limited the tumor immune-evasion [79]. These results and further reports encourage exploring this route in malignant gliomas [80,81]. To our knowledge, this modality is yet to be implemented in the context of malignant gliomas.

Nanotechnology-based delivery has emerged as a revolutionarily tool to transfer packed drugs into the target organ after systemic administration; minimalizing possible off-tumor side effects [82,83]. These nanotechnology-based carriers fortified with glioma-targeting peptides are capable to bypass the BBB due to the peptide-transport capacity of the BBB [84]. Such a nano-system has been widely investigated in malignant gliomas, particularly GBM [85]. However, the nanoparticle-aided CAR-T cell delivery in GBM is yet to be investigated. Interestingly, Säälik et al. 2019 could successfully employ a P32-directed tumor-penetrating peptide LinTT1 (KRGARSTA) to further enhance the GBM delivery of a systemically given abraxane nanoparticles (nano-formulated paclitaxel-albumin; Nab-paclitaxel). The investigators conjugated LinTT1 iron oxide nanoparticles (nano-worms; NWs) to the anti-GBM amphiphilic proapoptotic _D_(KLAKLAK)_2_ peptide in PDX mice and detected significant tumor responses [86]. This finding is in line with the earlier efforts by Agemy et al. 2011; using NWs coated with CGKRK_D_(KLAKLAK)_2_ [87]. Regarding CAR-T cells delivery in GBM, Xie et al. 2021 generated ultra-small superparamagnetic particles of iron oxide (USPIOs) glucose-coated nanoparticles to facilitate tracking the EGFRvIII- and IL13Rα2- CAR-T cells in human GBM xenografts. The authors demonstrated that CAR-T cells can encapsulate USPIO complexes without provoking changes in cell morphology or functionality, while allowing MRI tracking of the cell infiltration which indeed was detected and persist from day 14–21. Interestingly, the investigators detected a higher expression of the intracellular tight junction, VE-cadherin, in tumor-derived endothelial cells; indicating the restoration of intact BBB [88]. More studies are still needed to further establish this route of administration in malignant gliomas.

## 3. The Current Landscape of Clinical CAR-T Trials in Malignant Gliomas

In the recent years, the amount of clinical trials applying CAR-T cells in the context of malignant gliomas and particularly glioblastoma has dramatically expanded [31,34], with not only variable application routes and larger patient cohorts, but also novel CAR-T target antigens, augmented valency of applied CAR-Ts, co-therapies with immunomodulatory agents (including ICI), multiple time-point of repetitive therapies and various brain tumor entities included, including newly-diagnosed tumors.

Surface antigens tested in pre-clinical studies were almost all transferred to clinical testing, although the lion’s share of studies involves, among others, EGFRvIII (Table 2). Three of the studies concerning EGFRvIII have been terminated and one study involving a co-therapy with pembrolizumab was completed, with results pending. While the status of one study is unknown, two of the studies are still recruiting (NCT03423992 and NCT03170141). The BRITE study, expected to have started in May 2022, will test the intravenous application of hEGFRvIII-CD3-biscFv bispecific T cell engaged in conjunction with autologous T-cells (ACT) in malignant glioma. All studies on EGFRvIII chose intravenous application, with the exception NCT03283631, which uses intratumoral application via convection enhanced delivery (withdrawn), and NCT03170141, a phase I/II study with interventional arms involving intravenous and intratumoral application, making this study quite unique. CAR-T cells, EGFR806 CAR-T cells, which can potentially recognize tumor-specific untethered EGFR and EGFRvIII, are currently under testing in recurrent and refractory pediatric CNS tumors (NCT03638167) [89]. In this, particular, study, both an intracavitary and intraventricular route are being investigated. NCT05168423 is another study projected to start recruiting soon also involving EGFR, however as a bivalent CAR in combination with IL-13Ra2.

For IL-13Ra2, results of four trials are expected, with three trials that have been initiated in the past three years and that are currently recruiting. Each of these studies enrolls distinct patient populations: Recurrent glioblastoma (in combination with ICI), malignant pediatric brain tumors and a third study including both. Since published studies on IL-13Ra2 utilized locoregional application, all these studies apply CAR-Ts either into the tumor or intracerebroventricularly. For the antigen GD2, three phase I trials with mostly pediatric tumor entities are awaited and like the initial intravenous application in the trial by Majzner et al. 2022 [33], all involve an intravenous route.

HER2 CAR-Ts have experienced attention in terms of studies with glioblastoma patients. One phase I trial testing intravenous application of HER2 CMV-specific CAR-T cells has been completed, with data still unpublished. Three HER2 phase I studies are currently recruiting and include either malignant gliomas (grade 3 and 4) for application into the tumor cavity or, for the study including HER2+ CNS tumors in general, also an intraventricular application route. Interestingly, NCT03383978 uses irradiated off-the-shelf NK-92 as CAR recipients, that are injected intracavitarily.

B7-H3-targeted CAR-T cells are exploited in three studies. In two phase I studies, patients with either recurrent glioblastoma or, in this study with a large-planned patient cohort (90 patients—NCT04185038), patients with DIPG/DMG and recurrent or refractory pediatric CNS tumors are included. A further phase I/II B7-H3-CAR-T study on recurrent glioblastoma patients is expected to enroll its first patient in May 2022. All aforementioned studies apply CAR-Ts via intracavitary or intraventricular routes.

CD147, CD70 and chlorotoxin all represent further antigens for which CAR-T phase I trials have been conceptualized and initiated. A phase I trial with CAR-T cells against CD 147 in recurrent malignant gliomas has been recruiting since approximately three years. Another phase I study on MPP2-positive recurrent or progressive glioblastoma is set to determine the feasibly of applying chlorotoxin-derived CAR-Ts and in addition involves a study arm receiving a dual intratumoral and intraventricular application. NCT05353530 is phase I trial designed to test CD70-directed CAR-Ts applied intravenously after the end of radiotherapy in patients with newly diagnosed CD70-positive and MGMT-unmethylated glioblastoma, IDH wildtype.

## 4. Conclusions and Future Prospects

Effective CAR-T cell therapy in malignant gliomas requires innovative solutions to ensure efficient CAR-T cell delivery and efficacy while minimizing off-target or on-target of tumor toxicity. Available routes of administration include: intraventricular, intra-tumoral, intra-arterial and systemic i.v. injection, FUS-aided delivery, nanoparticle-guided delivery and biomaterial-aided delivery. Each route exhibits a variety of unique aspects that should be taken into consideration when conceptualizing and implementing clinical trials.

It can be speculated that successor CAR-T cell trials in malignant gliomas tend to utilize the same application route as in previous trials of the same target antigen. For example, EGFRvIII CAR-T trials were mostly intravenous, while IL-13Ra2-targeting trials predominantly make use of locoregional delivery, mostly by intracavitary injection. The decision for intracerebroventricular vs. intracavitary application should conceptually consider if the targeted tumor type is known to frequently spread into the CSF or in the ependymal niche [90]. The idea that a cerebroventricular application could provide a benefit for e.g., medulloblastoma or ependymoma or beyond these entities is entertained by the theory that a semi-systemic route of administration, i.e., an administration into a close but circulating system (such as the ventricles of the brain) could prove indeed more effective than mere intra-tumoral or intravenous therapy. The intra-tumoral delivery generates a regional reaction that might be too confined, whereas the intravenous one faces the challenge of homing to the brain and localizing into the tumor. Because malignant glioma is known to colonize throughout the whole brain, a semi-systemic application modality could potentially prove suitable. This is supported by the observation that in some pre-clinical models, intraventricular delivery of CAR-T cells led to the most pronounced therapeutic responses in tumor-bearing mice, surpassing both systemic intravenous delivery and direct injection into the tumor site [50,90,91]. Nevertheless, a systemic route of administration supported with delivery enhancers such as FUS and nanocarriers provides an attractive approach undergoing intensive preclinical investigations. To our knowledge, there are no trials directly comparing both intracranial and intravenous routes of application in the same CAR-T cell target antigen and tumor entity. Comprehensive clinical trials comparing these different routes in terms of efficacy and safety are needed to better elucidate CAR-T cell delivery in malignant gliomas.

Tumor antigen loss after the infiltration of CAR-T cells into the tumor site represents a common mechanism of immune escape and resistance to CAR-T cell therapy [16,90,92,93]. Therefore, designing multivalent CARs could contribute to the success of the therapy. A second mechanism of tumor escape is through the activation of the immune checkpoints to confer CAR-T cell exhaustion [94]. Hence, combining ICI with CAR-T cell therapy is expected to enhance the efficacy of the therapy. Interestingly, as more specifically, ex vivo CRISPR-based knockout of the immune checkpoints; e.g., PD-1, in the autologous or allogenic CAR-T cells is a promising approach to further enhance the efficacy of CAR-T cell therapy [95]. Interestingly, Wang et al. 2018 described an early exhaustion of IL13Rα2^+^-targeting CD8^+^ CAR-T cells upon stimulation with IL13Rα2^+^ GBM cells and in an orthotopic GBM model *in vivo*; compared to a persistent activation of CD4^+^ CAR-T cells [96]. The preserved CD4^+^ CAR-T cell activity was positively correlated to the recursive cytotoxicity of CAR-T cells [96]. Interestingly, Tang et al. 2022 demonstrated a positive correlation between the expression of PD-1—but not CTLA4, TIM3 nor LAG3—on EGFRvIII-targeting CD4^+^ CAR-T cells, with the area under the curve of log10 CAR copies/mg genomic DNA in peripheral blood post-infusion, and the PFS in patients with EGFRvIII-expressing recurrent GBM [97]. Another mechanism of glioma resistance to CAR-T cell therapy is the aberrant death receptor signaling in tumor cells which makes them insensitive to CAR-T cell-induced apoptosis [98,99,100]. Concomitant administration of targeted drugs such as PI3K-ß inhibitor, bcl-2 family apoptosis inhibitor ABT-737 or histone deacetylase inhibitors such as LBH589 was evinced to regulate tumor apoptosis and enhance the sensitivity of resistant cancer cells in preclinical studies [100,101,102].

Constructing CARs on other immune cells is becoming an emerging field which involves the innate immunity into front-line anti-tumor response. CAR natural killer (NK) cells are thought to produce less side effects compared to CAR-T cells, presumably due to their lower tendency to release cytokine storm and hence cause fewer cases of CRS and neurotoxicity. In addition, the short life-span of NK cells in the circulation diminish the risk of on-target/off-tumor toxicity, which on the other hand, could possibly render them less effective [103]. Most recently, The CAR-based therapy has expanded and encompassed macrophages. Indeed, CD19-targeted THP-1 macrophage encoding the intracellular domain CD3ζ showed high antigen-specific phagocytosis against CD19^+^ K562 cancer cells in vitro [104]. Furthermore, HER2-targeted CAR macrophages modulate the tumor microenvironment in vivo by dominating the M1-like phenotype and improved the overall survival [104]. Considering the fact that macrophages are tissue-resident by nature, this might help in better adapting of the cells to the locoregional administration and contribute to the success of the therapy. Further studies are needed to further investigate the feasibility of these approaches.

In conclusion, pre-clinical and clinical studies on the application of CAR-T cells in malignant gliomas have demonstrated safety and potential anti-tumor efficacy, though currently constrained by challenges in immunoediting such as CAR-T cell activation and antigen depletion. Furthermore, as clinical trials on different antigens still assess the preclinically more favored intracranial application route vs. intravenous application of CAR-T cells, innovative approaches of delivery such as focused ultra-sound and nanoparticles are yet to find their way into clinical translation.

Box 1T cells in GBM microenvironment [105,106,107].Tumor-infiltrating lymphocytes represent 0.25% of cells in human GBM [87]. They include mostly CD4^+^ T helper, CD4^+^CD25^+^FoxP3^+^ regulatory T cells (Treg) and CD8^+^ T cytotoxic cells. By secreting CCL2, GBM cells recruit Treg and induce Treg-mediated T cell inhibition through several cytokines; including TIM4 and IDO secretion. Indeed, CD8^+^ T cells isolated from GBM samples were shown to be exhausted with an elevated expression of co-inhibitory immune checkpoints PD-1, CTLA-4, TIM-3, LAG-3, CD39, VISTA, BTLA, 2B4, CD160 and TIGIT. The interaction between PD-1 on T cells and PD-L1 on GBM and myeloid cells, in addition to the competition between CTLA-4 and T cell activating receptor CD28 for B71 and B72 on APC, produce inhibitory signals to exhaust T cells. The majority of CD4^+^ T identified in GBM tissue are Treg cells which are functionally immunosuppressive or functionally inactive. GBM cells reserve their “cold” tumor phenotype by attracting Treg, myeloid-derived suppressor cells (MDSCs) and regulatory dendritic cells that halt the activity of T cells, while Treg and MDSCs suppress NK cells activity.

Box 2CAR-T cells and CAR-T generations [108,109].Chimeric antigen receptors (CARs) are synthetic receptors engineered to harbor an antibody-derived extracellular ligand-binding domain, a hinging transmembrane domain and an intracellular T cell receptor (TCR)-derived signaling domain fortified with other co-stimulatory domains. Comprising antibody-derived variable regions, CAR-T cells are capable of skipping major histocompatibility complex (MHC) expression and presentation by tumor cells or professional antigen presenting cells (APC) and directly recognize extracellular domains and proteins on, for example, tumor cells (tumor antigens). The natural ligands of cellular surface receptors can also be modified and hence utilized as extracellular recognition domains.Tampering with the intracellular signaling domain and adding further co-stimulatory signal effectors to these synthetic receptors has led to the development of second, third, fourth and fifth-generation CARs. Because first-generation CARs lacked co-stimulatory signaling domains, they showed limited efficacy. Therefore, second-generation and further CAR designs have included co-stimulatory domains to enhance T cell activation signaling and render it more durable, whereas the third-generation encompasses multiple co-stimulation domains. The fourth-generation was designed as T-cells redirected for universal cytokine-mediated killing which have augmented cytotoxicity by expressing transgenic cytokines in the tumor microenvironment. While the fifth generation of CAR-T-cells are multivalent in addition to other cytokine-expressing domains

## Figures and Tables

**Figure 1 biomedicines-10-01738-f001:**
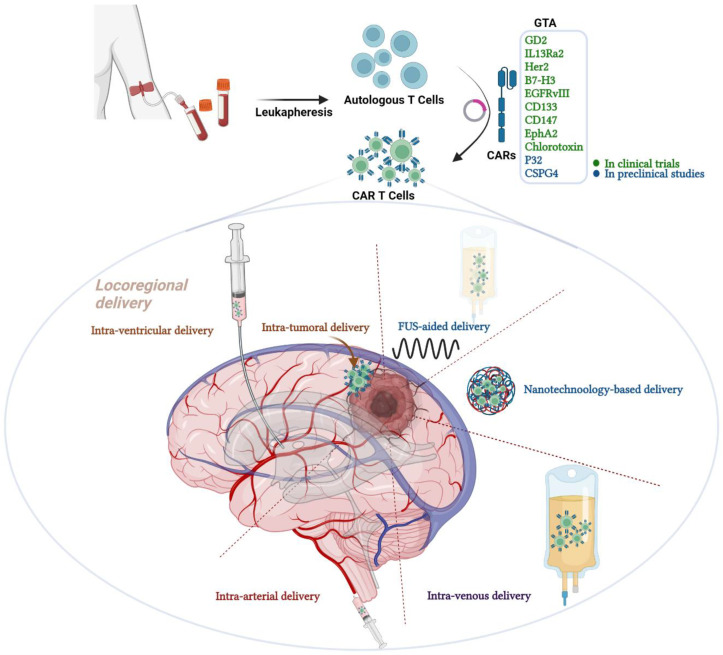
Graphical abstract representing the different routes of chimeric antigen receptor (CAR) T cell therapy administration in malignant gliomas. CAR T cell therapy is one of the immunotherapeutic modalities under the umbrella of adoptive T cell therapy. CAR T cells are genetically modified autologous T cells that carry CARs to better recognize and attack cancer cells. The construct composed of antibody-derived extracellular ligand-binding domain, a hinging transmembrane domain and an intracellular T cell receptor (TCR)-derived signaling domain fortified with other co-stimulatory domains. Examples of GTAs that are targeted in clinical trials are GD2, IL13Ra2, HER2, B7-H3, EGFRvIII, EphA2 and chlorotoxin. Other GTAs that are under preclinical investigations include P32 and CSPG4. Several routes of administration are available for CAR T cell therapy but vary in the efficacy and safety; including locoregional delivery, FUS-aided delivery, nanotechnology-based delivery, intravenous and intra-arterial delivery. GTA; glioma target antigens, CARs; chimeric antigen receptors, GD2; disialoganglioside 2, IL13Ra2; interleukin 13 receptor subunit alpha 2, HER2; Human epidermal growth factor receptor 2, B7-H3; B7 Homolog 3, EGFRvIII; epidermal growth factor receptor variant III, EphA2; Ephrin type-A receptor 2 and FUS; Focused Ultrasound. Created with BioRender.com.

**Table 2 biomedicines-10-01738-t002:** Ongoing clinical trials involving CAR-T cell therapies in malignant gliomas.

Clinical Trial ID	Trial Start Month.JJ	Phase	Tumor Entity	CAR	Application	Subject	Status	Notes
** *Locoregional delivery of CAR-T cells in glioma* **
NCT02208362	May 15	I	Recurrent or refractory malignant glioma	IL13 Rα2	Intracavitary Intraventricular	82	Active, not recruiting	
NCT02442297	May 15	I	Glioblastoma	HER2	Intratumoral, Intracavitary Intraventricular	28	Recruiting	
NCT03170141	May 17	I/II	Glioblastoma	EGFRvIII (Original), Current: many targets	Intravenous Intratumoral	20	Enrolling by invitation	Lymphodepleting with Fludarabine and Cyclophosphamide
NCT03383978	Dec. 17	I	Recurrent HER2-positive glioblastoma	HER2 Natural Killer (NK) CAR	Intracranial	30	Recruiting	Also intraoperative injections
NCT03283631	May 18	I	EGFRvIII+ recurrent glioblastoma	EGFRvIII	Intratumoral via convection enhanced delivery	2	Terminated	Stereotactic radiosurgery followed by infusion of CAR-T cells
NCT03500991	Jul. 18	I	Recurrent or refractory HER2-positive CNS tumors	HER2	Intracavitary Intraventricular	48	Recruiting	First results published (Vitanza et al. 2021)
NCT03389230	Aug. 18	I	Recurrent or refractory grade 3–4 glioma	HER2	Intratumoral Intracavitary	42	Recruiting	Study arm with dual intratumoral and intraventricular application
NCT03638167	Mar. 19	I	EGFR-positive recurrent or refractory pediatric CNS tumors	EGFR806	Intracavitary Intraventricular	36	Recruiting	Application route depends on tumor localization
NCT04045847	May 19	I	Recurrent malignant glioma	CD147	Intracavitary Intratumoral	31	Recruiting	
NCT04003649	Sep. 19	I	Recurrent or refractory glioblastoma	IL13 Rα2 +/− nivolumab and ipilimumab	Intracavitary Intraventricular	60	Recruiting	
NCT04185038	Dec. 19	I	DIPG/DMG and recurrent or refractory pediatric CNS tumors	B7-H3	Intracavitary Intraventricular	90	Recruiting	DIPG.(Diffuse Intrinsic Pontine Gliomas): ventricular applicationNon-DIPG: Tumor cavity
NCT04214392	Jan. 20	I	MPP2-positive recurrent or progressive glioblastoma	Chlorotoxin-derived	Intratumoral, Intracavitary Intraventricular	36	Recruiting	Study arm with dual intratumoral and intraventricular application
NCT04385173	May 20	I	Recurrent and refractory glioblastoma	B7-H3-targeted CAR-T cells with temozolomide	Intratumoral Intraventricular	12	Recruiting	Application within temozolomide cycles
NCT04510051	Dec. 20	I	Recurrent or refractory pediatric CNS tumors	IL13Rα2	Intraventricular	18	Recruiting	Lymphodepleting with Fludarabine and Cyclophosphamide
NCT04661384	May 21	I	Leptomeningeal glioblastoma, ependymoma, or medulloblastoma	IL13 Rα2	Intraventricular	30	Recruiting	hinge-optimized 41BB-co-stimulatory CAR truncated CD19-expressing
NCT04077866	May 22	I/II	Recurrent or refractory glioblastoma	B7-H3-targeted CAR-T cells with or without temozolomide	Intracavitary Intraventricular	40	Recruiting	Application within temozolomide cycles
** *Systemic delivery* **
NCT01109095	Oct. 10	I	Recurrent or refractory glioblastoma	HER2 CMV-specific CAR-T cells	Intravenous	16	Completed	
NCT02209376	Nov. 14	I	EGFRVIII+ Glioblastoma	EGFRvIII	Intravenous	11	Terminated	Sponsor decision to terminate prior to completion to pursue combination therapies
NCT02575261	Oct. 15	I/II	EphA2-positive malignant glioma	EphA2	Intravenous	0	Withdrawn	
NCT02664363	Jul. 16	I	Primary glioblastoma	EGFRvIII	Intravenous	3	Terminated	Leukapheresis pre-therapy Study funding ended
NCT02844062	Jul. 16	I	Glioblastoma	EGFRvIII	Intravenous	20	Recruiting/Unknown	Lymphodepleting with Fludarabine and Cyclophosphamide
NCT0293844	Jul. 16	I	Recurrent glioblastoma multiforme	PD-L1	Intravenous	20	Recruiting/Unknown	Lymphodepleting with Fludarabine and Cyclophosphamide switch receptor modified
NCT03423992	Mar. 18	I	Recurrent malignant gliomas	EGFRvIII, IL13Rα2, Her-2, CD133, EphA2, GD2	Intravenous	100	Recruiting	Different antigens
NCT03726515	Mar. 19	I	Newly diagnosed MGMT-unmethylated glioblastoma	EGFRvIII + pembrolizumab	Intravenous	7	Completed	MGMT-Unmethylated Glioblastoma
NCT04099797	Feb. 20	I	GD2-positive brain tumors	GD2	Intravenous	34	Recruiting	Lymphodepleting with Fludarabine and Cyclophosphamide
NCT04196413	Jun. 20	I	Diffuse Intrinsic Pontine Gliomas (DIPG) & Spinal Diffuse Midline Glioma(DMG)	GD2	Intravenous	54	Recruiting	Lymphodepleting with Fludarabine and Cyclophosphamide
NCT04903795	May. 22	I	Grade 4 malignant glioma	hEGFRvIII-CD3-biscFv Bispecific T cell engager (BRiTE)	Intravenous	18	No recruitment yet	with and without peripheral autologous T-cell (ACT) infusion
NCT05298995	May. 22	I	refractory pediatric CNS tumors	GD2	Intravenous	54	Not yet recruiting	Lymphodepletion with conventional chemotherapeutics
NCT05168423	Jun. 22	I	EGFR-amplified glioblastoma	EGFR-IL13Ra2	Intravenous	18	Not yet recruiting	Lymphodepleting with Fludarabine and Cyclophosphamide
NCT05353530	Jun. 22	I	Newly diagnosed CD70 positive and MGMT-unmethylated adult glioblastoma	CD70	Intravenous	18	Not yet recruiting	CAR-T therapy after end of radiotherapy

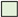
 EGFRvIII, 
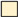
 IL13 Rα2, 
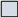
 HER2, 
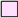
 CD147, 
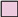
 B3-H7, 
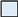
 Chlorotoxin-derived, 
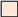
 EphA2, 
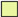
 CD70.

## Data Availability

Not applicable.

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
