# Peer review of "Site-Specific Considerations on Engineered T Cells for Malignant Gliomas"

_biomedicines, 2022, doi:10.3390/biomedicines10071738_

Round 1
Reviewer 1 Report
This manuscript gives a good overview of CAR-T therapy in glioma. The major targets that are in pre-clinical and clinical development are discussed extensively. The linguistic style is very good, which makes the manuscript easy to read and understand. All the latest relevant literature is cited.
There are some minor issues to be addressed before publication.
Table 2. It would be more clear if the authors could classify them according to their combination or absence of other therapies (such as chemotherapy, immune checkpoint inhibitors…)
Table 2, please indicate the year of “Trial start”
It would be helpful if the authors include more studies on strategies to overcome the tumor resistance to CAR-T cells, like, how to improve CAR-T infiltration and persistence in the tumor microenvironment, CAR-T cell efficacy.
Author Response
Referee 1:
This manuscript gives a good overview of CAR-T therapy in glioma. The major targets that are in pre-clinical and clinical development are discussed extensively. The linguistic style is very good, which makes the manuscript easy to read and understand. All the latest relevant literature is cited.
We thank the referee for this very positive comment.
There are some minor issues to be addressed before publication.
Table 2. It would be more clear if the authors could classify them according to their combination or absence of other therapies (such as chemotherapy, immune checkpoint inhibitors…)
The authors intended to classify the trials according to the route of administration and the CAR T cell targeted antigens to fit well the scope of the review. Any adjuvant therapy is now included in the column (Note) of the table.
Table 2, please indicate the year of “Trial start”
The “trial start” is already included in table 2. The order of hierarchy is now edited and the trial start is listed in column 2. In addition to the route of administration, the trials are now classified chronologically.
It would be helpful if the authors include more studies on strategies to overcome the tumor resistance to CAR-T cells, like, how to improve CAR-T infiltration and persistence in the tumor microenvironment, CAR-T cell efficacy.
We thank the referee for this important comment and added to the section “future prospects”.
Reviewer 2 Report
The review article " Site-specific considerations on engineered T cells for malignant gliomas ", presented by Nirmeen Elmadany et al., is a very well presented and methodic review of the new immunotherapy strategies on glioblastoma.
The way the authors introduce and explain the different techniques and strategies are very comprehensible, being also very well accompanied by clear graphics. I really think is a fantastic review.
Therefore, I have only a minor comment.
I see that you only cited trials developed or registered at NIH (USA and Canada). Have you searched in the European, the Asian, the Australian & New Zealand, and African clinical trials registers?
I think it will increase the number of assays (not much to be honest), but enhance the power of you great review.
Author Response
The review article " Site-specific considerations on engineered T cells for malignant gliomas ", presented by Nirmeen Elmadany et al., is a very well presented and methodic review of the new immunotherapy strategies on glioblastoma.
The way the authors introduce and explain the different techniques and strategies are very comprehensible, being also very well accompanied by clear graphics. I really think is a fantastic review.
Therefore, I have only a minor comment.
We thank the referee for this fantastic feedback.
I see that you only cited trials developed or registered at NIH (USA and Canada). Have you searched in the European, the Asian, the Australian & New Zealand, and African clinical trials registers? I think it will increase the number of assays (not much to be honest), but enhance the power of you great review.
The studies included in table 2 are in part from China. In addition, the authors had a look on the website for European trials registrations with all queries e.g., CAR-T / CAR / glioma / GBM render 0 results.
https://www.clinicaltrialsregister.eu/ctr-search/search?query=T+cell+glioma
Moreover, the authors checked the Chinese database with no further results.
http://www.chictr.org.cn/searchprojen.aspx?title=&officialname=Glioblastoma&subjectid=&secondaryid=&applier=&studyleader=ðicalcommitteesanction=&sponsor=&studyailment=&studyailmentcode=&studytype=0&studystage=0&studydesign=0&minstudyexecutetime=&maxstudyexecutetime=&recruitmentstatus=0&gender=0&agreetosign=&secsponsor=®no=®status=0&country=&province=&city=&institution=&institutionlevel=&measure=&intercode=&sourceofspends=&createyear=0&isuploadrf=&whetherpublic=&btngo=btn&verifycode=1VTB6&page=4
Reviewer 3 Report
The development of CAR T cell technology is critical for the treatment of a number of malignancies, including glioblastoma. Here, the authors present a timely and good overview of the role of CAR T cells, including ways to introduce CAR T cells, results of preclinical studies and planned treatment protocols. The relevance of this publication is not in doubt. Some shortcomings that could be pointed out include a lack of originality (for example, a similar review was published in Cancers in 2021 doi: 10.3390/cancers13215445, but it has a more in-depth analysis of the information, which this manuscript lacks). Nevertheless, the article is quite well written and could be published in Biomedicines.
Author Response
The development of CAR T cell technology is critical for the treatment of a number of malignancies, including glioblastoma. Here, the authors present a timely and good overview of the role of CAR T cells, including ways to introduce CAR T cells, results of preclinical studies and planned treatment protocols. The relevance of this publication is not in doubt. Some shortcomings that could be pointed out include a lack of originality (for example, a similar review was published in Cancers in 2021 doi: 10.3390/cancers13215445, but it has a more in-depth analysis of the information, which this manuscript lacks). Nevertheless, the article is quite well written and could be published in Biomedicines.
Thank you very much for this positive feedback.